# Primary versus Secondary Elevations in Fundus Autofluorescence

**DOI:** 10.3390/ijms241512327

**Published:** 2023-08-02

**Authors:** Rait Parmann, Stephen H. Tsang, Janet R. Sparrow

**Affiliations:** 1Departments of Ophthalmology, Columbia University, 635 W. 165th Street, New York, NY 10032, USA; 2Departments of Pathology and Cell Biology, Columbia University, 635 W. 165th Street, New York, NY 10032, USA

**Keywords:** fundus autofluorescence, quantitative fundus autofluorescence, short-wavelength fundus autofluorescence, optical coherence tomography, retina

## Abstract

The method of quantitative fundus autofluorescence (qAF) can be used to assess the levels of bisretinoids in retinal pigment epithelium (RPE) cells so as to aid the interpretation and management of a variety of retinal conditions. In this review, we focused on seven retinal diseases to highlight the possible pathways to increased fundus autofluorescence. *ABCA4*- and *RDH12*-associated diseases benefit from known mechanisms whereby gene malfunctioning leads to elevated bisretinoid levels in RPE cells. On the other hand, *peripherin2/RDS*-associated disease (*PRPH2/RDS*), retinitis pigmentosa (RP), central serous chorioretinopathy (CSC), acute zonal occult outer retinopathy (AZOOR), and *ceramide kinase like* (*CERKL*)-associated retinal degeneration all express abnormally high fundus autofluorescence levels without a demonstrated pathophysiological pathway for bisretinoid elevation. We suggest that, while a known link from gene mutation to increased production of bisretinoids (as in *ABCA4*- and *RDH12*-associated diseases) causes primary elevation in fundus autofluorescence, a secondary autofluorescence elevation also exists, where an impairment and degeneration of photoreceptor cells by various causes leads to an increase in bisretinoid levels in RPE cells.

## 1. Introduction

Autofluorescence (AF) is the innate ability of fluorophores to emit light after having absorbed photons of suitable wavelength. The retina exhibits an autofluorescence [1,2,3], the source of which is the bisretinoid lipofuscin [1,2,4]. Bisretinoids forms non-enzymatically in the outer segments of photoreceptor cells due to random reactions of vitamin A aldehyde with phosphatidylethanolamine [5,6,7]. This complex mixture of fluorophores is transferred secondarily to retinal pigment epithelium (RPE) within the phagocytosed outer segment membrane, where they accumulate in lysosomes with age [8,9]. In vivo measurement of retinal AF was first described by Delori [1,8] using a fundus spectrophotometer. The techniques of noninvasive AF measurement have since advanced and, in clinical settings, a confocal scanning laser ophthalmoscopy (cSLO) is widely used [10,11]. Lipofuscin can be excited with short wavelength (SW) light of 440–545 nm wavelength [5] and it has a wide emission band peaking at approximately 620–630 nm [8]. For short wavelength fundus autofluorescence (SW-AF) imaging, the cSLO uses an argon laser generating 488 nm (blue) excitation light. The instrument registers the emitted light with a detector and a barrier filter that transmits light from 500–680 nm to reduce out of focus light and to minimize the contribution of AF from the crystalline lens [8,12,13].

Pathological increases in bisretinoid formation are a primary feature of diseases associated with deficiencies in ABCA4 and RDH12. Under the latter conditions, these by-products of the visual cycle are understood to form as a direct result of the dysfunction conferred by the disease-causing gene variants. Here, we propose that elevated bisretinoid can also be a secondary feature of some retinal diseases when accelerated bisretinoid formation occurs downstream of photoreceptor cell impairment and degeneration.

## 2. Quantitative Fundus Autofluorescence

To compare SW-AF images obtained from different subjects at specific retinal locations, or from the same subject longitudinally, a standardized method is applied. For quantitative fundus autofluorescence (qAF), a scanning laser ophthalmoscope (cSLO) is employed with an excitation light of 488 nm, and emitted light is captured within a range of 500 to 680 nm [14]. Protocols have been established for image acquisition and a formula provides calculations of qAF from grey levels [14,15]. This method overcomes differences in sensitivity settings and laser power by the use of an internal reference mounted within the instrument; the reference normalizes the AF from the fundus to the fluorescence of the standard [14]. Additionally, the acquisition protocol has limitations for sensitivity settings and only non-normalized images (without histogram stretch that is otherwise utilized to enhance image quality but makes grey levels incomparable between images) are saved and analyzed. The formula used for calculations takes into account corneal curvature and refractive errors, and is only applicable to phakic eyes. Image acquisition is performed with a 30° × 30° field lens and a dilated pupil (>7 mm) in a darkened room. Photoreceptor bleaching for 20 s precedes imaging to reduce the absorption by photopigment, which can act as a screen to exciting and emitted light [14,16]. Acquired images are analyzed using custom software. Gray level values are calculated within predetermined regions—three concentric rings each divided into eight segments (Figure 1) [14,17]. Primary use is made of qAF values calculated for areas situated 7–9° eccentric from the fovea (the middle ring). Macular pigment (lutein and zeaxanthin) concentrated in the fovea has an absorption range (400–540 nm) similar to that of lipofuscin but levels are negligible at an eccentricity of 7° and beyond [18,19,20]. In healthy subjects, qAF values increase with age because of lipofuscin accumulation [21]. qAF values are higher in Caucasians versus African Americans and Asians, and in females and smokers. Spatially, qAF is highest superotemporally [21]. Procedures for measuring fundus autofluorescence using 488 nm excitation (short wavelength fundus autofluorescence) in both humans [22] and mice have been developed [23,24].

## 3. Retinal Diseases Exhibiting Elevated Fundus Autofluorescence

### 3.1. ABCA4-Associated Disease

One of the disorders most characterized by elevated bisretinoid lipofuscin is *ABCA4*-associated disease. Indeed, elevated lipofuscin is considered to be a key feature and early component of the *ABCA4*-disease process [25,26,27]. Disease-causing variants in *ABCA4* are the leading cause of inherited childhood or adolescence macular degeneration [28,29]. Progressive central vision loss, color vision defects and photophobia are the most common symptoms observed in the first two decades [30]. The hallmark findings in the fundus are AF flecks, central chorioretinal atrophy, and peripapillary sparing (Figure 2) [31,32,33,34,35]. Early childhood *ABCA4*-associated disease often evolves to cone-rod dystrophy with panretinal degeneration and substantial vision loss [36,37,38]. Late-onset disease is typically associated with the mild end of the severity spectrum, often with preserved visual acuity because of foveal spearing [39]. Another *ABCA4*-associated disease phenotype is the bull’s eye maculopathy characterized by disease restricted to central retina [40,41].

*ABCA4* disease is caused by bi-allelic variants in the gene that encodes an ATP-binding cassette (ABC) transporter in photoreceptor outer segments that transports N-retinylidene-phosphatidylethanolamine (NRPE), a reversible Schiff base adduct of retinaldehyde and phosphatidylethanolamine [42,43,44,45,46]. This activity aids in the reduction of 11-cis- and all-trans-retinaldehyde to retinol. Under conditions of ABCA4-deficiency, NRPE is more available to react with a second retinaldehyde molecule, leading to the non-reversible formation of bisretinoids such as A2-GPE (A2-glycerophosphoethanolamine), all-trans-retinal dimer, A2E and cis-isomers of A2E, and A2-DHP-PE (A2-dihydropyridine-phosphatidylethanolamine) [47,48,49,50]. A variety of approaches, from histology [26,51] to the use of imaging modalities, have shown that eyes with *ABCA4* disease present with higher levels of lipofuscin compared to healthy age-matched eyes (Figure 2D) [8,17,52]. Prior to the use of qAF, fundus AF was measured spectrophotometrically at a position 7° temporal to the fovea using an excitation wavelength of 510 nm [53]. Under this protocol, autofluorescence intensity was found to be approximately three-fold higher in patients having *ABCA4*-associated disease relative to control subjects of comparable age. Consistent with this earlier study, qAF levels were also shown to be elevated in *ABCA4*-associated disease [17]. While it is challenging to extract a phenotype–genotype correlation in *ABCA4* disease due to the large numbers of identified disease-causing variants and the frequency of compound heterozygous mutations, the mutation L2027F and the complex allele L541P/A1038V conferred relatively high levels of qAF in young patients, while in patients carrying homozygous mutations in G1961E, qAF was within normal limits for age [17]. It is notable that, in the latter group of patients, the disease presents as a bull’s eye maculopathy without disease features (flecks, atrophy) 7–9° outside the fovea. Given that the qAF approach normalizes fundus grey levels to an internal fluorescence reference, in the future, these groups of patients can also be studied longitudinally.

An elegantly designed alternative approach to measuring short wavelength fundus autofluorescence utilizes reduced illuminance, short wavelength excitation with postacquisition image processing (SW-RAFI) rather than the higher conventional AF imaging intensities [54]. In patients presenting with *ABCA4*-associated disease, the investigators quantified intensities along horizontal profiles through the fovea and correlated these measurements with structural information from optical coherence tomography (OCT) scans and functional data recorded by microperimetry. They observed hyperautofluorescence in the macula that, at different disease stages, was associated with and without abnormalities in visual function. Unexpectedly, they also found that, in some cases, hyperautofluorescence with SW excitation was accompanied by increased near-infrared autofluorescence, the predominant source of which is melanin. This finding was later replicated [55].

Accumulated bisretinoids are responsible for the cellular atrophy that accompanies disease-causing *ABCA4* variants in humans and *Abca4*-null mutations in mice [56,57,58,59]. Bisretinoids are cytotoxic in large measure because they are amphiphilic molecules, they exhibit photoreactivity [57,60,61,62,63,64], and they can be acted upon by oxidizing products of Fenton chemistry [65,66].

### 3.2. RDH12-Associated Retinal Disease

Retinol dehydrogenase enzymes in photoreceptor cells are members of a family of short-chain dehydrogenases that execute NADPH-dependent reduction of retinaldehyde to retinol. This reducing activity is crucial due to the reactivity of free retinaldehyde. To this end, these enzymes use NADPH to reduce both 11-cis-retinaldehyde and all-trans-retinaldehyde [67,68,69,70,71]. Retinol dehydrogenase 8 (RDH8) has been localized to outer segments and, in mice, RDH8 deficiency causes a mild phenotype exhibiting delayed dark adaptation [72].

*Retinol dehydrogenase 12* is expressed in inner segments of rods and cones [73,74].

Null mutation in *Rdh12* confers increased susceptibility to retinal light injury [74], a finding consistent with bisretinoid-associated photo damage [75]. Combined deletion of both *Rdh8* and *Rdh12* in mice leads to increased A2E measurable at 3 months of age; this is followed by progressive rod-cone dystrophy [73,76,77]. In murine rods, a reduction of all-trans-retinal may require the activity of both *Rdh8* and *Rdh12* [78].

In the presence of biallelic recessive mutations in *RDH12*, human retinal disease presents with varying age of onset and phenotypic severity. Specifically, coding variants in *RDH12* can bestow severe disease with onset in early childhood (2–4 years). *RDH12* mutations account for approximately 4% of autosomal recessive Leber congenital amaurosis (LCA). Features of macular atrophy include loss of RPE as indicated by fundus hypoautofluorescence, hypertransmission of spectral domain optical coherence tomography (SD-OCT) signal, and outer retinal tubulations (Figure 3) [69,70,79,80,81,82]. Peripapillary sparing can also be observed [83]. Other characteristics of the fundus, such as intraretinal bone spicule pigmentation [69,83], are typical of a retinitis pigmentosa phenotype.

While for *ABCA4*- and *RDH12*-associated disease the pathway from gene variant to accelerated bisretinoid formation and elevated SW-AF has been demonstrated, there are retinal pathologies where higher AF values, representing elevated bisretinoid levels, are measured without a known mechanism explaining the accentuated bisretinoid lipofuscin.

### 3.3. Peripherin2/RDS-Associated Disease

*Peripherin2/RDS*-associated disease (*PRPH2/RDS*) is generally associated with an autosomal-dominant inheritance and is known to cause retinitis pigmentosa (RP), several forms of macular dystrophy, and cone-rod dystrophies [84]. The phenotypes vary from macular and peripheral atrophy to flecks and vitelliform material [85,86,87]. The gene encodes a photoreceptor specific protein located primarily in the rim region of the outer segment disc and lamellae [88] and is thought to play an essential role in disc formation, stabilization, and maintenance [89,90]. It is proposed that mutations in the *PRPH2* gene lead to major structural abnormalities of the outer segment, ultimately resulting in loss of visual function and photoreceptor degeneration [91]. Acquisition of SW-AF images has proven to be the most adequate method for early disease recognition, and it is also the only method capable of distinguishing between different patterns of the disease [92]. Considerable phenotypic overlap (Figure 2B,D and Figure 4B,E) can exist between *ABCA4* and *PRPH2/RDS*-associated disease [93]; yet current pathophysiological understanding of the dysfunctional PRPH2 protein does not predict a concomitant increase in bisretinoid production. By plotting qAF values as a function of age, a cross-sectional study has revealed increased SW-AF intensities in eyes with *PRPH2/RDS*-associated disease compared to healthy age-matched controls [94]. This increase is, however, lower than in patients harboring *ABCA4* disease (Figure 4C).

### 3.4. Retinitis Pigmentosa

Retinitis pigmentosa refers to a diverse group of hereditary retinal disorders in which abnormalities of photoreceptor cells lead to progressive vision loss. Over 100 RP-causing genes, explaining only 40–50% of all RP patients, have been identified to date [95,96]. Many of the associated genes encode proteins involved in phototransduction, the visual cycle, photoreceptor structure, or gene transcription, yet in some cases the gene function is poorly understood or remains unknown [97,98]. In most cases of RP there is an initial degeneration of the rods followed by degeneration of cones [99]. Although RP is genetically and phenotypically heterogeneous, its clinical presentation often includes pigmented deposits (bone spicules), resulting from proliferating and migrating RPE cells, vascular attenuation, and waxy pallor of optic disc [100,101,102]. Another sign often present is a hyperAF ring or arc in SW-AF images; these rings can constrict over time (Figure 5E) [103,104,105,106,107]. The ring marks a transition from preserved photoreceptor ellipsoid zone (EZ) (inner border of the ring) to disrupted EZ and thinning of the outer nuclear layer (ONL) (within the ring), to absence of EZ and thinning or absence of ONL (outer border of the ring) [108,109,110]. By applying the qAF protocol to analyze the hyperAF rings, we found that in some patients (28% of the cohort) the qAF values were increased compared to corresponding locations in a healthy retina (Figure 5C,D) [111]. We considered factors that could account for the visibility of SW-AF rings in RP. The qAF imaging protocol includes photopigment bleaching before image acquisition, thus the unmasking of the SW-AF is unlikely to contribute to an explanation for the elevated autofluorescence. The abnormal autofluorescence cannot be attributed to accelerated phagocytosis of photoreceptor outer segments by RPE since the bisretinoid fluorophores responsible for SW-AF form in photoreceptor cells prior to phagocytosis. If thinning of the overlying neural retina with the creation of a window defect was an explanation for the SW-AF in RP, one would expect AF within the ring to be higher at all times, but this was not observed. Instead, we suggest that there is an increase in the formation of bisretinoid lipofuscin resulting in elevated SW-AF. Considering that the RP cohort discussed here included individuals that differed in age, genotype, and clinical stage, it is reasonable to presume that increased qAF is present at some stage of the disease in most hyperAF rings.

HyperAF rings in RP are not genotype-specific [112,113,114] and are observed in a majority of RP patients [115]. Some forms of autosomal dominant RP are attributable to opsin mutations that confer inherent instability in the Schiff base linkage between 11-cis-retinal and the opsin protein [116,117,118,119,120] even in the dark; this instability forces 11-cis-retinal to isomerize to the all-trans-configuration and to leave the binding pocket in opsin [121]. Perhaps this is one mechanism that can serve as an explanation for unchecked retinaldehyde, elevated bisretinod formation, and hyperAF. These mutations are also notable given that light is considered to accentuate the disease [122].

### 3.5. Central Serous Chorioretinopathy

Central serous chorioretinopathy (CSC) is clinically characterized as detachment of neurosensory retina secondary to leakage through RPE [123,124]. Its etiology is unknown and it pathophysiology is not well understood, but choroidal hyperpermeability, RPE, and hormonal pathways seem to play a key role [125,126]. Apart from several identified risk factors like corticosteroid usage, Cushing syndrome, pregnancy, and type A personality, genetic predisposition seems to play an important role indicated by frequent familial cases [125,127,128,129] with *complement factor H* gene (*CFH*) being associated in numerous studies [130,131,132]. *CFH* plays a significant role in RPE-choroid complex as an inhibitor of the alternate pathway; it also causes vasodilatation of the choroidal vessels and increases microvascular permeability, indicating a possible association with CSC pathogenesis [133,134]. In SW-AF images, CSC presents in a variety of patterns that can be associated with chronicity, visual acuity, and the integrity of the EZ line [135]. Although the classic qAF_8_ ring analysis did not indicate a generalized relationship between SW-AF and CSC, all CSC lesions were associated with changed SW-AF topography and localized deviated qAF intensities (Figure 6C) [136]. For instance, the advancing front of the CSC lesion had elevated qAF levels; it also disrupted EZ and thinned ONL, and hyperreflective debris in the outer retina were associated with abnormally increased qAF [136]. In CSC, there is no known direct pathway to elevated levels of lipofuscin, thus elevated levels of qAF might be associated with processes of photoreceptor cell degeneration and secondary increases in bisretinoid formation [136].

### 3.6. Acute Zonal Occult Outer Retinopathy

Acute zonal occult outer retinopathy (AZOOR) is a rare condition of non-genetic origin that affects predominantly young women and is characterized by acute onset, photopsias, and subjective visual field losses [137,138]. Based on multiple imaging modalities, the primary damage has been identified at the level of the photoreceptor outer segments, and choriocapillaris can be involved secondarily as a collateral damage [139,140,141,142]. The pathogenic mechanism remains unknown, although a viral or autoimmune etiology has been proposed [143,144]. In SW-AF images, AZOOR commonly presents as diffuse patches of hyperautofluorescence outside the central macula, along with a peripapillary area of abnormal AF delimited by a border of high AF (AZOOR line). In OCT scans, photoreceptor cell-attributable layers are abnormal with disruptions of the ellipsoid band and interdigitation zone [139,145]. In a study in which qAF was analyzed at the transition zone, SW-AF intensity was elevated in three out of six patients [146]. It was also observed that SW-AF levels can be impermanent at the transition zone (AZOOR line), meaning that AF values comparable to those of healthy eyes may have been higher at an earlier stage of the disease. Based on a current understanding of photoreceptors being at the center of disease processes in AZOOR, the elevated qAF levels in the transition zone accompanied by EZ loss and ONL thinning indicate that photoreceptor cell degeneration may involve increased bisretinoid formation. It could be significant that patients with lesion borders delineated by a hyperautofluorescent AZOOR line are reported to more frequently undergo disease progression [139].

### 3.7. Ceramide Kinase like-Associated Retinal Degeneration

Mutations in the *ceramide kinase-like* (*CERKL*) gene have been reported to cause autosomal recessive RP and cone-rod dystrophy [147,148]. Clinical presentation involves early-onset maculopathy with severe generalized retinal dysfunction, peripheral lacunae, and hyperAF foci on SW-AF (Figure 7D) [149]. Current knowledge of the *CERKL* gene indicates it has protective functions for photoreceptor cells against oxidative stress through several pathways [150,151,152,153]. Thus, mutations in that gene would be expected to cause oxidative damage leading to photoreceptor cell death and retinal degeneration [154]. So far, only a handful of patients have been studied with the qAF protocols but the results indicate increased macular qAF levels in regions of advanced disease associated with RPE and photoreceptor degeneration (Figure 7C) [149]. Indeed, five out of six patients described in the literature have demonstrated borderline or substantially higher qAF values [149,155]. These results, along with the hyperAF foci in SW-AF images, support the theory of secondarily increased AF caused by disabled and/or degenerating photoreceptor cells.

## 4. Discussion

Following capture of a photon of light, the 11-cis-retinaldehye chromophore of visual pigment isomerizes to all-trans-retinaldehyde, thereby initiating a series of conformational rearrangements leading to the phototransduction cascade. To sustain vision, 11-cis-retinaldehye also has to be re-formed via the visual cycle [156]. Because it bears a reactive aldehyde, all-trans-retinaldehyde must first be reduced to all-trans-retinol by NADPH-dependent retinol dehydrogenases (RDHs) in the photoreceptor cell (RDH8 and RDH12) [42,73,76,77,157,158].

Some of the retinaldehyde released into the photoreceptor disk membrane is immediately accessible to the RDH enzymes; alternatively, all-trans-retinal and excessive 11-cis-retinal can undergo condensation reactions with the primary amine of phosphatidylethanolamine (PE) in the outer segment disc membrane thereby forming the adduct N-retinylidene-PE (NRPE) via a Schiff base linkage (C=C-N) [159]. NRPE is the ligand that binds the photoreceptor-specific ATP-binding cassette transporter (ABCA4) in outer segments [42,44,45,160,161,162,163]. The function of ABCA4 is to transport NRPE across the lipid bilayer to the cytoplasmic face of the disc membrane, where NRPE hydrolyzes and all-trans-retinaldehyde is released and reduced to the less reactive alcohol (all-trans-retinol) by the NADPH-dependent retinol dehydrogenases [42,73,158,164]. The formation of NRPE likely serves to shepherd vitamin A aldehyde and to guard against acute aldehyde injury. However, inefficient clearance of NRPE can result in its reaction with a second retinaldehyde and irreversible formation of toxic bisretinoids. Inability of the photoreceptor cell to execute any of these energy-consuming processes due to a variety of disease-causing factors may lead to adverse bisretinoid production.

## Figures and Tables

**Figure 1 ijms-24-12327-f001:**
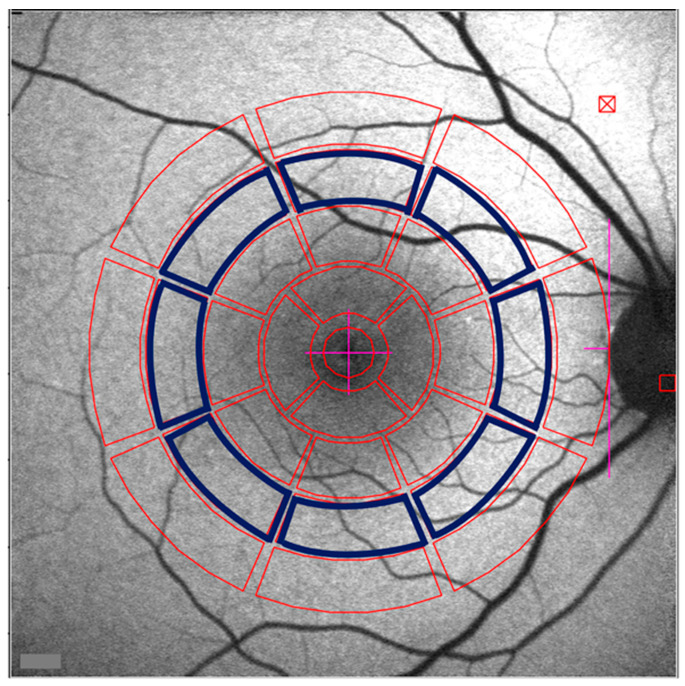
Quantitative measurement of fundus autofluorescence (qAF). To obtain an average qAF value, mean gray levels are recorded from eight circularly arranged segments (blue outlines) at 7–8° eccentricity, centered on fovea and limited by the edge of optic disk.

**Figure 2 ijms-24-12327-f002:**
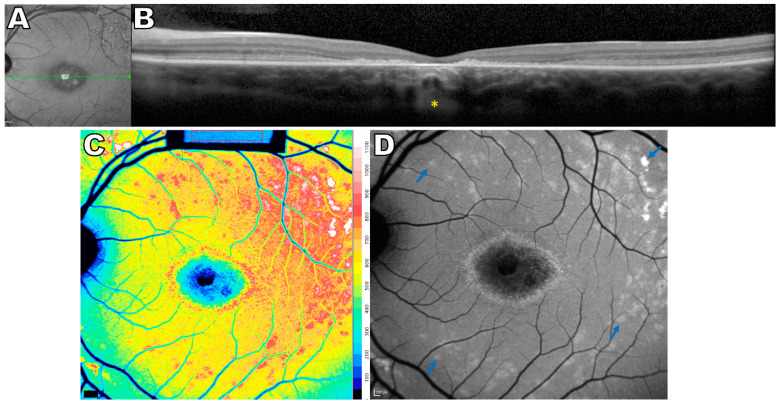
Multimodal images of *ABCA4*-associated disease, 11 year old male. (**A**) Near-infrared reflectance image showing the axis (green line) of the spectral domain optical coherence tomography scan (SD-OCT). (**B**) SD-OCT scan demonstrates central outer retinal loss and increased hypertransmission into the choroid (yellow asterisk) (**C**) Quantitative fundus autofluorescence (qAF) color-coded image exhibiting generalized increase of AF. Color scale of qAF units (0–1200) is provided on the right margin in C. (**D**) Central atrophy and hyper-autofluorescent flecks (blue arrows) are visible in the short wavelength fundus autofluorescence image. The figure is modified from the paper [35].

**Figure 3 ijms-24-12327-f003:**
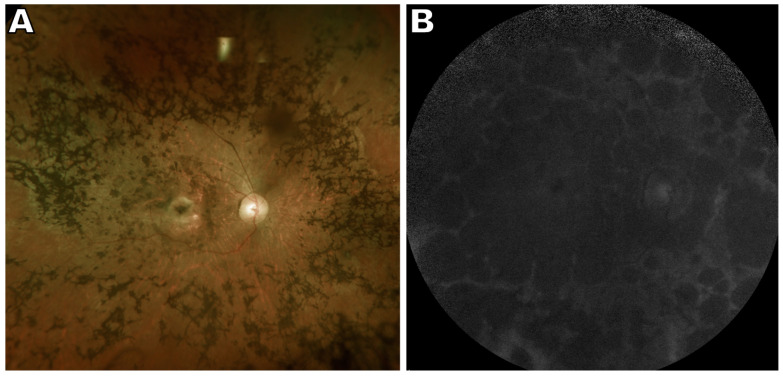
Multimodal images of *RDH12*-associated retinal disease, 34 year old male. (**A**) Ultrawide-field pseudocolor fundus image reveals atrophic macula, optic disc pallor, and extensive intraretinal pigment migration. (**B**) Widespread retinal atrophy indicated by fundus hypoautofluorescence in short wavelength fundus autofluorescence image. The figure is modified from the paper [79].

**Figure 4 ijms-24-12327-f004:**
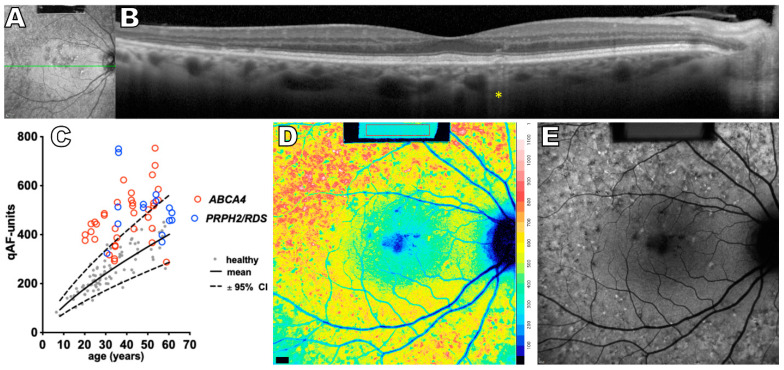
Multimodal images of *peripherin2/RDS*-associated disease, 53 year old female. (**A**) Near-infrared reflectance image showing the axis (green line) of the spectral domain optical coherence tomography scan (SD-OCT). (**B**) SD-OCT scan exhibiting temporal outer retinal loss, increased hypertransmission into the choroid (yellow asterisk) and a fleck. (**C**) Quantitative fundus autofluorescence (qAF) values of PRPH2/RDS-associated disease (blue circles) and ABCA4-associated disease (red circles) plotted as a function of age. Mean (solid line) and 95% confidence intervals (dashed lines) are shown for white, healthy eyes (gray filled circles). (**D**) qAF color-coded image. Elevated short wavelength fundus autofluorescence (SW-AF) levels can be appreciated. Color scale of qAF units (0–1200) is provided on the right margin in (**D**). (**E**) SW-AF image presents a minor central hypoautofluorescence and numerous flecks. The figure is modified from the paper [94].

**Figure 5 ijms-24-12327-f005:**
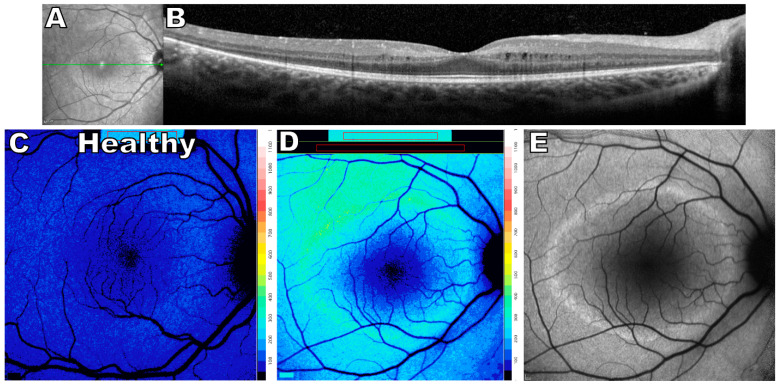
Multimodal images of retinitis pigmentosa, 10 year old female. (**A**) Near-infrared reflectance image showing the axis (green line) of the spectral domain optical coherence tomography scan (SD-OCT). (**B**) SD-OCT scan exhibits loss of external limiting membrane, ellipsoid zone, and outer nuclear layer temporally and nasally. Cystoid macular edema is visible in parafoveal region (**C**) Quantitative fundus autofluorescence (qAF) color-coded image of a healthy eye age-matched to the patient in D. (**D**) Quantitative fundus autofluorescence color-coded image displaying elevated SW-AF levels spatially correlating to the hyper autofluorescent ring visible in image (**E**). Color scale of qAF units (0–1200) is provided on the right margin. (**E**) SW-AF image displaying the wide hyperautofluorescent ring. The figure is modified from the paper [111].

**Figure 6 ijms-24-12327-f006:**
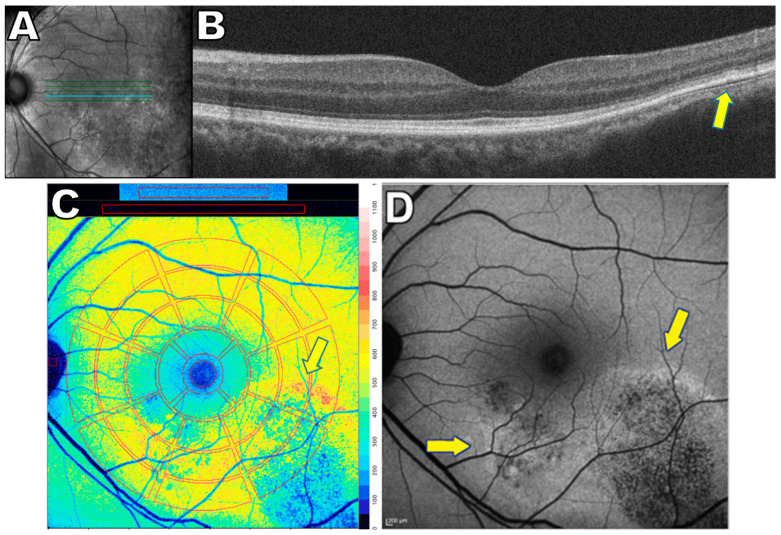
Multimodal images of central serous chorioretinopathy, 48 year old male. (**A**) Near-infrared reflectance image showing the axis (bold cyan line) of the spectral domain optical coherence tomography scan (SD-OCT). (**B**) SD-OCT scan reveals minimal pigment epithelium detachment and disorganized ellipsoid zone temporally. This correlates (yellow arrows) with elevated quantitative fundus autofluorescence (qAF) and hyperautofluorescent signal in short wavelength fundus autofluorescence (SW-AF) images (respectively **C** and **D**). (**C**) qAF color-coded image with an overlapping qAF grid (red lines). In infero-temporal retina, a decrease in qAF is visible (blue pseudocolor) with a surround of an increased qAF (red pseudocolor). Color scale of qAF units (0–1200) is provided on the right margin. (**D**) SW-AF image reveals that the center of the lesion is hypoautofluorescent while the margins are hyperautofluorescent (yellow arrows). The figure is modified from the paper [136].

**Figure 7 ijms-24-12327-f007:**
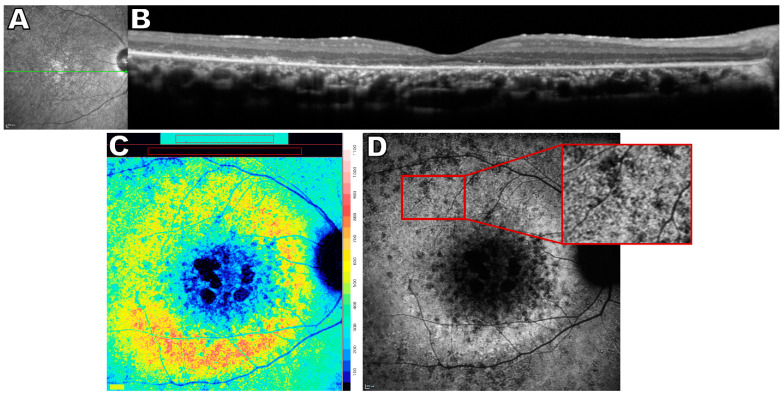
Multimodal images of *ceramide kinase-like*-associated retinal degeneration, 24 year old male. (**A**) Near-infrared reflectance image showing the axis (green line) of the spectral domain optical coherence tomography scan (SD-OCT). (**B**) Chorioretinal degeneration and peripheral thinning is visible in the SD-OCT scan. (**C**) Quantitative fundus autofluorescence (qAF) color-coded image demonstrates centrally decreased qAF surrounded by prominently increased qAF levels. Color scale of qAF units (0–1200) is provided on the right margin. (**D**) Short wavelength fundus autofluorescence image exhibits a dense pattern of hyperautofluorescent foci in region adjacent to atrophy. The inset on the right is a magnification of the region indicated on the left. The figure is modified from the paper [149].

## Data Availability

Not applicable.

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
