# Peer review of "Primary versus Secondary Elevations in Fundus Autofluorescence"

_ijms, 2023, doi:10.3390/ijms241512327_

Round 1

Reviewer 1 Report

The authors provided a description of the application of qAF in primary and secondary AF alterations. It provided some information of the AF imaging features in retinal diseases, and meanwhile I have some concerns and hope the authors to address.

Line 73-78

Please provide a schematic figure of the quantitative protocol (predetermined regions, the three concentric rings, and the divided eight segments.

Line 87-92, 119-123, 177

As is known and as mentioned by the authors, there is significant heterogeneity of retinal degenerations. It is sensible to compare qAF longitudinally, while I believe it should take great caution to get conclusion when it comes to comparison between genotypes and even between different entities of diseases (as illustrated in Line 119-123, Line 177). Moreover, qAF may also associated with duration of the disease, so comparison should take disease duration into consideration.

Reviewer 2 Report

This is a really nice review paper by Parman et al about primary versus secondary elevations in fundus autofluorescence. It comprises essential results and techniques about especially quantative autofluorescence. Retinal diseases and their corresponding appearance are adressed adeuqately. 

However, while the authors include the benefit of of qAF, they may also consider to mentiuon and discuss e.g. the potential detrimental effect of AF due to photo-oxidation and the role of lipofuscin. As a consequence, e.g. the ProgStar study group the approach of SW-RAFI. This potnetial detrimental effect should be at least shortly discussed.

Otherwise it is a very well written and illustrative manuscript.

Round 2

Reviewer 1 Report

The authors has made adequate revision of the manuscript according to the comments given by the reviewer. I would suggest accepting the manuscript.

Reviewer 2 Report

All my comments have been adequately addressed.